# Hydrothermal plumes as hotspots for deep-ocean heterotrophic microbial biomass production

Cécile Cathalot [1✉], Erwan G. Roussel [2], Antoine Perhirin[3], Vanessa Creff[2], Jean-Pierre Donval[1], Vivien Guyader [1], Guillaume Roullet[4], Jonathan Gula [4,5], Christian Tamburini[6], Marc Garel[6], Anne Godfroy[2] & Pierre-Marie Sarradin [3]

Carbon budgets of hydrothermal plumes result from the balance between carbon sinks through plume chemoautotrophic processes and carbon release via microbial respiration. However, the lack of comprehensive analysis of the metabolic processes and biomass production rates hinders an accurate estimate of their contribution to the deep ocean carbon cycle. Here, we use a biogeochemical model to estimate the autotrophic and heterotrophic production rates of microbial communities in hydrothermal plumes and validate it with in situ data. We show how substrate limitation might prevent net chemolithoautotrophic production in hydrothermal plumes. Elevated prokaryotic heterotrophic production rates (up to 0.9 $gCm^{-2}y^{-1}$) compared to the surrounding seawater could lead to 0.05 $GtCy^{-1}$ of C-biomass produced through chemoorganotrophy within hydrothermal plumes, similar to the Particulate Organic Carbon (POC) export fluxes reported in the deep ocean. We conclude that hydrothermal plumes must be accounted for as significant deep sources of POC in ocean carbon budgets.

[1] Laboratoire Cycles Géochimiques et ressources – LCG/GM/REM, Ifremer, Plouzané, France. [2] Laboratoire de Microbiologie des Environnements Extrêmes – LMEE/EEP/REM, Ifremer, Plouzané, France. [3] Laboratoire Environnement Profond – LEP/EEP/REM, IFREMER, Plouzané, France. [4] Univ Brest, CNRS, IRD, Ifremer, Laboratoire d'Océanographie Physique et Spatiale (LOPS), IUEM, Plouzané, France. [5] Institut Universitaire de France (IUF), Paris, France. [6] Aix-Marseille Univ, Université de Toulon, CNRS, IRD, MIO UM110 Marseille, France. ✉email: cecile.cathalot@ifremer.fr

A major challenge in understanding the oceanic carbon cycle and forecasting the impact of climate change at a global scale is quantifying the portion of organic carbon escaping the upper water column and being sequestered in the deep ocean. Models of water-column attenuation, deep-sea carbon flux, and sequestration rates in the global carbon budgets are estimated from net primary production and export fluxes derived from satellite-imaging of ocean surface. However, misbalance between total biological carbon demand (heterotrophic production and respiration) and particulate organic carbon (POC) flux[1,2], export to the deep ocean through dissolved organic carbon (DOC)[3] or local fertilization by spermwhales[4] seriously challenge such approach. Episodic organic carbon pulses have only recently been highlighted as significant contributors to the total POC flux[5], pointing out temporal and spatial variations of the carbon pump efficiency.

Deep-sea hydrothermal venting is a global process occurring where fluid circulation in the oceanic crust is close to a heat source, especially along mid-ocean ridges (MOR) and back-arc basins. Hydrothermal vents play a significant role in heat and matter exchange between the Earth's interior and the hydrosphere, such as export sources of dissolved and particulate metals that could drive ocean biogeochemical cycles at a global scale[6,7]. Part of the imbalance in the global carbon budgets[2] has been suggested to originate from fertilization of surface waters by hydrothermally-derived iron[8]. Despite our growing understanding on chemical metal scavenging and export processes within hydrothermal plumes, their contribution to the deep-sea carbon flux still remains elusive[9]. Hydrothermal vents along mid-ocean ridge systems host unique, highly productive communities mainly relying on chemoautotrophic primary production. Although it has recently been shown that hydrothermal subseafloor communities are highly productive[10], microbial communities growth rates, carbon biomass production rates, and associated POC export fluxes within hydrothermal non-buoyant plumes remain enigmatic[9].

Numerous numerical models based on thermodynamics mixing approaches have been used to estimate dominant microbial metabolism in hydrothermal chimney walls and plumes by calculating the amount of energy potentially available through a wide range of pathways[11–13]. Two studies have incorporated a kinetic component allowing to spatially resolve the geochemical reactions and metagenomics within the hydrothermal plume or chimney[14,15]. Here, we couple these biogeochemical numerical approaches describing chemical species distribution in hydrothermal plumes with a microbial model based on the Microbial Transition State (MTS) theory of growth[16,17] to predict microbial cell distribution and carbon fixation rates within the plumes. Chemoorganotrophic and chemolithoautotrophic[18] carbon biomass production rates in various hydrothermal plumes along slow, ultra-slow, and fast-spreading ridges were then compared with rates measured in situ.

Here, we show that by promoting microbial biomass production through chemoorganotrophy along the thousands of kilometers of MOR, hydrothermal plumes process a significant fraction of the deep ocean carbon pool. Our results suggest that common assumptions of vertical flux attenuation used in global carbon model are strongly underestimated, as chemoorganotrophic biomass production rates below 1000 m are at least twice the hypothesized POC flux export values.

## Results and discussion
### Biogeochemical modeling of biomass microbial production in hydrothermal plume.
In our numerical model of hydrothermal plume, we consider 27 pertinent chemical species and 18 relevant microbial metabolisms[19,20] including catabolic and anabolic processes (Table 1), and an uniform first order mortality rate[15] of $1.16 \times 10^{-8}\,\mathrm{s}^{-1}$. We apply our model to ten hydrothermal sites in various MOR contexts (slow, intermediate, and fast-spreading rates) over a wide range of both hydrodynamic and chemical conditions. Non-buoyant plume chemistry is predicted based only on dilution from high-temperature fluids with seawater, ignoring thus potential abiotic processes that could occur and alter the chemical distribution in the rising plume[21,22]. As a result, concentrations of some substrates may be slightly overestimated (Supplementary Table 3). However, as the rising speed of a hydrothermal buoyant plume is usually quite fast[23], the short residence times (between 1.4 and 2 h for all sites considered) would only lead to a slight overestimation of substrate concentration and affect only total sulfides and $H_2$ concentrations[22]. Other biogeochemical processes (e.g., oxidation, adsorption) are not significantly affected as they occur over longer timescales, within the neutral buoyant plume[24]. To prevent any overestimation of substrates in NBP leading to miscellaneous biomass production rates, we compared our NBP predicted values to literature data and corrected them accordingly when needed (Supplementary Table 3). The distribution of DOC in the plume could only be higher than the theoretical dilution with seawater by in situ plume production or lateral entrainment of high DOC diffused flows[25]. A simple spatially resolved hydrodynamic model based on the original buoying jet equations set up by Morton and Turner[26,27] is used to predict the maximal plume height and associated dilution ratios. When hydrodynamic information is missing (e.g., fluid output velocity, fluid vent diameter), we consider a dilution factor of $10^5$ respective to concentrations in the high-temperature fluids[28]. The distribution of the different microbial groups is hypothesized based on literature[15,29] and taken identical through all study sites. Microbial functional community model is based on the MTS conceptual framework[16,17], including water and ionic activities[30]. Initial cell densities of $10^5$ cells ml$^{-1}$ considered in our model were derived from cell counts performed within TAG hydrothermal plume during the BICOSE 2 cruise and was consistent with previously reported values[31]. The average residence time for the microbial community (30 days) that was chosen to run the model was long enough to allow for consumption of initial chemical species, microbial turnover, and internal recycling (i.e., labile DOC release)[32,33].

### Chemolithoautotrophic rates in hydrothermal plumes.
Our modeling results suggest that, for all 10 hydrothermal sites studied, chemical distribution in non-buoyant plumes is unable to support net microbial chemolithoautotrophic biomass production. Substrate concentrations in the plume appear too low to harvest the threshold energy level needed in our MTS model to sustain the chemolithoautotrophic microbial community (i.e., trigger microbial cell division)[17]. Such substrate limitation in deep open ocean waters is not surprising as the concentrations of electron donors or acceptors required to trigger pelagic chemolithoautotrophy activity usually occur in surface waters, or in zonal redox gradient areas such as Oxygen Minimum Zones[34,35]. Only very low dark chemolithoautotrophic rates around $0.005\,\mu\mathrm{C\,l^{-1}\,d^{-1}}$ have been reported in bathy- and meso-pelagic seawaters (below 1000 m depth) mainly through aerobic ammonia and nitrite oxidation processes[36–38]. Given the formalism of our model, reproducing such low rates in deep Atlantic seawater implies to both increase the $V_{\mathrm{harv}}$ parameter and decrease the dissipated free energy of growth $\Delta G_{\mathrm{diss}}$ for chemolithoautotrophic processes down to the acetate heterotrophy levels. This basically comes down to increasing the energy harvested through catabolism or, in other words, decreasing the energy threshold needed

**Table 1 Metabolic pathways included in the model.**

| | Catabolism | Anabolism |
|---|---|---|
| **Chemolithoautotrophic processes:** | | |
| 1- Sulfide oxidation: | $HS^- + 0.5O_2 + H^+ \rightarrow S + H_2O$ | $2.1HS^- + HCO_3^- + 0.2NH_4^+ + 2.9H^+ \rightarrow 2.1S + 2.5H_2O + CH_{1.8}O_{0.5}N_{0.2}$ |
| 2- Sulfur oxidation: | $S + 1.5O_2 + H_2O \rightarrow SO_4^{2-} + 2H^+$ | $0.7S + HCO_3^- + 0.2NH_4^+ + 0.3H_2O \rightarrow 0.7SO_4^{2-} + 0.6H^+ + CH_{1.8}O_{0.5}N_{0.2}$ |
| 3- Thiosulfate oxidation: | $S_2O_3^{2-} + 2O_2 + H_2O \rightarrow 2SO_4^{2-} + 2H^+$ | $0.525S_2O_3^{2-} + HCO_3^- + 0.2NH_4^+ + 0.125H_2O \rightarrow 1.05SO_4^{2-} + 0.25H^+ + CH_{1.8}O_{0.5}N_{0.2}$ |
| 4- Methane oxidation: | $CH_4 + 2O_2 \rightarrow HCO_3^- + H^+ + H_2O$ | $0.525CH_4 + HCO_3^- + 0.2NH_4^+ + 0.8H^+ \rightarrow 0.525CO_2 + 1.45H_2O + CH_{1.8}O_{0.5}N_{0.2}$ |
| 5- Hydrogen oxidation (Knallgas): | $H_2 + 0.5O_2 \rightarrow H_2O$ | $2.1H_2 + HCO_3^- + 0.2NH_4^+ + 0.8H^+ \rightarrow 2.5H_2O + CH_{1.8}O_{0.5}N_{0.2}$ |
| 6- Nitrification— step1 (nitritation): | $NH_4^+ + 1.5O_2 \rightarrow NO_2^- + 2H^+ + H_2O$ | $0.9NH_4^+ + HCO_3^- \rightarrow 0.6H^+ + 1.1H_2O + CH_{1.8}O_{0.5}N_{0.2}$ |
| 7- Nitrification— step2 (nitratation): | $NO_2^- + 0.5O_2 \rightarrow NO_3^-$ | $2.1NO_2^- + HCO_3^- + 0.2NH_4^+ + 0.8H^+ \rightarrow 2.1NO_3^- + 0.4H_2O + CH_{1.8}O_{0.5}N_{0.2}$ |
| 8- Iron oxidation: | $Fe^{2+} + 0.25O_2 + 1.5H_2O \rightarrow FeOOH + 2H^+$ | $4.2Fe^{2+} + HCO_3^- + 0.2NH_4^+ + 5H^+ \rightarrow 4.2Fe^{3+} + 2.5H_2O + CH_{1.8}O_{0.5}N_{0.2}$ |
| 9- Manganese oxidation[a]: | $Mn^{2+} + 0.5O_2 + H_2O \rightarrow MnO_2 + 2H^+$ | $Mn^{2+} + HCO_3^- + 0.2NH_4^+ + 1.7H_2O \rightarrow 2.1MnO_2 + 3.4H^+ + CH_{1.8}O_{0.5}N_{0.2}$ |
| **Chemolithoautotrophic coupled processes based on ref. [18]** | | |
| 10- Sulfide oxidation to sulfur coupled to denitrification: | $HS^- + 1.4H^+ + 0.4NO_3^- \rightarrow S + 1.2H_2O + 0.2N_2$ | $2.1HS^- + HCO_3^- + 0.2NH_4^+ + 2.9H^+ \rightarrow 2.1S + 2.5H_2O + CH_{1.8}O_{0.5}N_{0.2}$ |
| 11- Sulfide oxidation to sulfur coupled to DNRA: | $HS^- + 1.5H^+ + 0.25NO_3^- \rightarrow S + 0.75H_2O + 0.25NH_4^+$ | $2.1HS^- + HCO_3^- + 0.2NH_4^+ + 2.9H^+ \rightarrow 2.1S + 2.5H_2O + CH_{1.8}O_{0.5}N_{0.2}$ |
| 12- Hydrogen oxidation coupled to denitrification: | $H_2 + 0.4H^+ + 0.4NO_3^- \rightarrow 1.2H_2O + 0.2N_2$ | $2.1H_2 + HCO_3^- + 0.2NH_4^+ + 0.8H^+ \rightarrow 2.5H_2O + CH_{1.8}O_{0.5}N_{0.2}$ |
| 13- Hydrogen oxidation coupled to DNRA: | $H_2 + 0.5H^+ + 0.25NO_3^- \rightarrow 0.75H_2O + 0.25NH_4^+$ | $2.1H_2 + HCO_3^- + 0.2NH_4^+ + 0.8H^+ \rightarrow 2.5H_2O + CH_{1.8}O_{0.5}N_{0.2}$ |
| **Chemolithoautotrophic processes based on ref. [19]** | | |
| 14- Hydrogenotrophic methanogenesis: | $H_2 + 0.25HCO_3^- + 0.25H^+ \rightarrow 0.25CH_4 + 0.75H_2O$ | $2.1H_2 + HCO_3^- + 0.2NH_4^+ + 0.8H^+ \rightarrow 2.5H_2O + CH_{1.8}O_{0.5}N_{0.2}$ |
| 15- Hydrogenotrophic sulfate reduction: | $H_2 + 0.25SO_4^{2-} + 0.25H^+ \rightarrow 0.25HS^- + H_2O$ | $2.1H_2 + HCO_3^- + 0.2NH_4^+ + 0.8H^+ \rightarrow 2.5H_2O + CH_{1.8}O_{0.5}N_{0.2}$ |
| 16- Anaerobic oxidation of methane: | $CH_4 + SO_4^{2-} \rightarrow HCO_3^- + HS^- + H_2O$ | $0.525CH_4 + HCO_3^- + 0.2NH_4^+ + 0.8H^+ \rightarrow 0.525CO_2 + 1.45H_2O + CH_{1.8}O_{0.5}N_{0.2}$ |
| **Heterotrophic/Chemoorganotrophic processes acetate-based** | | |
| 17- Aerobic respiration: | $acetate + 2O_2 \rightarrow H^+ + 2HCO_3^-$ | $0.525acetate + 0.2NH_4^+ + 0.275H^+ \rightarrow 0.05HCO_3^- + 0.4H_2O + CH_{1.8}O_{0.5}N_{0.2}$ |
| 18- Denitrification: | $acetate + 1.6NO_3^- + 0.6H^+ \rightarrow 0.8N_2 + 2HCO_3^- + 0.8H_2O$ | $0.525acetate + 0.2NH_4^+ + 0.275H^+ \rightarrow 0.05HCO_3^- + 0.4H_2O + CH_{1.8}O_{0.5}N_{0.2}$ |

[a]Not found in the CHNOSZ thermodatabase. $\Delta_rG_0$ based on ref. [61].

for the chemolithoautotrophs to start cell division by increasing the volume of substrate accessible around each microbial cell and lowering the amount of energy dissipated through anabolism. Minimization of energy losses during anabolic and catabolic reactions reflects the ability of microbial communities to survive even while starving, and to respond to fresh food supply as has been shown in bathypelagic communities[39].

Although there are no direct in situ measurements of chemolithoautrophic production rates (defined hereafter as Dissolved Inorganic Carbon (DIC) assimilation rates) available in deep hydrothermal plumes to our knowledge, significant ammonia and methane oxidation rates have been reported along the Juan de Fuca ridge[40–42], and variable chemolithoautotrophic rates have been observed in a shallow water hydrothermal plume offshore Taiwan[43]. To gain further confidence in our model and its ability to give quantitative estimates of biomass production rates including through chemolithoautotrophic metabolisms, we ran it for hydrothermal plume sites where either substrate conditions might not be limiting[44] or chemolithoautotrophic activity rates were available[10,41,42]. For conditions reproducing the 2011 El Hierro eruptive event, where large amount of reduced chemical species were released in the water column, our model predicts very little chemolithoautotrophic production ($0.32\,\mu gC\,L^{-1}\,d^{-1}$, Fig. 1), which is consistent with in situ observation[45]. At the Endeavor site, we ran our model for 300 days to increase chemolithoautotrophic biomass production due to internal recycling and microbial turnover. However, model outputs predict methane oxidation rates up to $1.3\,nmol\,m^{-3}\,d^{-1}$ but no net chemolithoautotrophic biomass production in the Endeavor hydrothermal plume (Fig. S3). Our model was also ran for the Crab Spa vent site (East Pacific Rise) where recent work has provided estimates of chemolithoautotrophic productivity from incubation at in situ pressure and temperature[10]. Our model does predict stimulated chemolithoautotrophic activity and reproduce the metabolic pathways of hydrogen oxidation coupled and uncoupled to dissimilatory nitrate reduction to ammonium and denitrification (Fig. 1). However, predicted chemolithoautotrophic biomass production rates at the Crab Spa vent field ($0.7–5.8\,\mu gC\,L^{-1}\,d^{-1}$, Fig. 1) or chemolithoautotrophic activity rates modeled for the Endeavor Main Field are about ten folds lower than the chemolithoautotrophic rates reported by the authors[10,41,42]. As for seawater, lowering the energy-dissipating barrier (by increasing $V_{harv}$ and lowering $\Delta G_{diss}$) allows the model to increase significantly the chemolithoautotrophic biomass production rates by a factor 2, closer to the measured rates. Within hydrothermal chemolithoautotrophic communities, energy expenditures required to fix DIC differ among metabolic pathways and C-fixation cycle at stake: Calvin–Benson–Bassham, reductive tricarboxylic acid, or reductive acetyl coenzyme A[46]. Studies suggest that chemolithoautotrophic hydrothermal microbial communities have the ability to switch between available catabolic pathways to adopt the less energetically expensive DIC fixation pathway available given the thermodynamic constraints of the cost of biomass synthesis[16]. $\Delta G_{diss}$ values in our model are fixed per metabolism and based on the definition by Heijnen and Van Dijken[47]: individual $\Delta G_{diss}$ differ between metabolism and the C-fixation cycle considered for ATP synthesis, they include the energy costly reverse transport of electron for numerous chemolithoautotrophic processes and are therefore likely to be overestimated. The reductive acetyl-CoA pathway is, for instance, not considered although it may be used by chemolithoautotrophs inhabiting $H_2$-rich environments, where $CO_2$ is directly reduced by $H_2$ and ATP input is therefore not required[46,47]. However, this level of metabolic consideration is beyond the scope of this paper, and our modeling approach lays the groundwork for estimating chemolithoautotrophic biomass production rates in seawater and hydrothermal plumes.

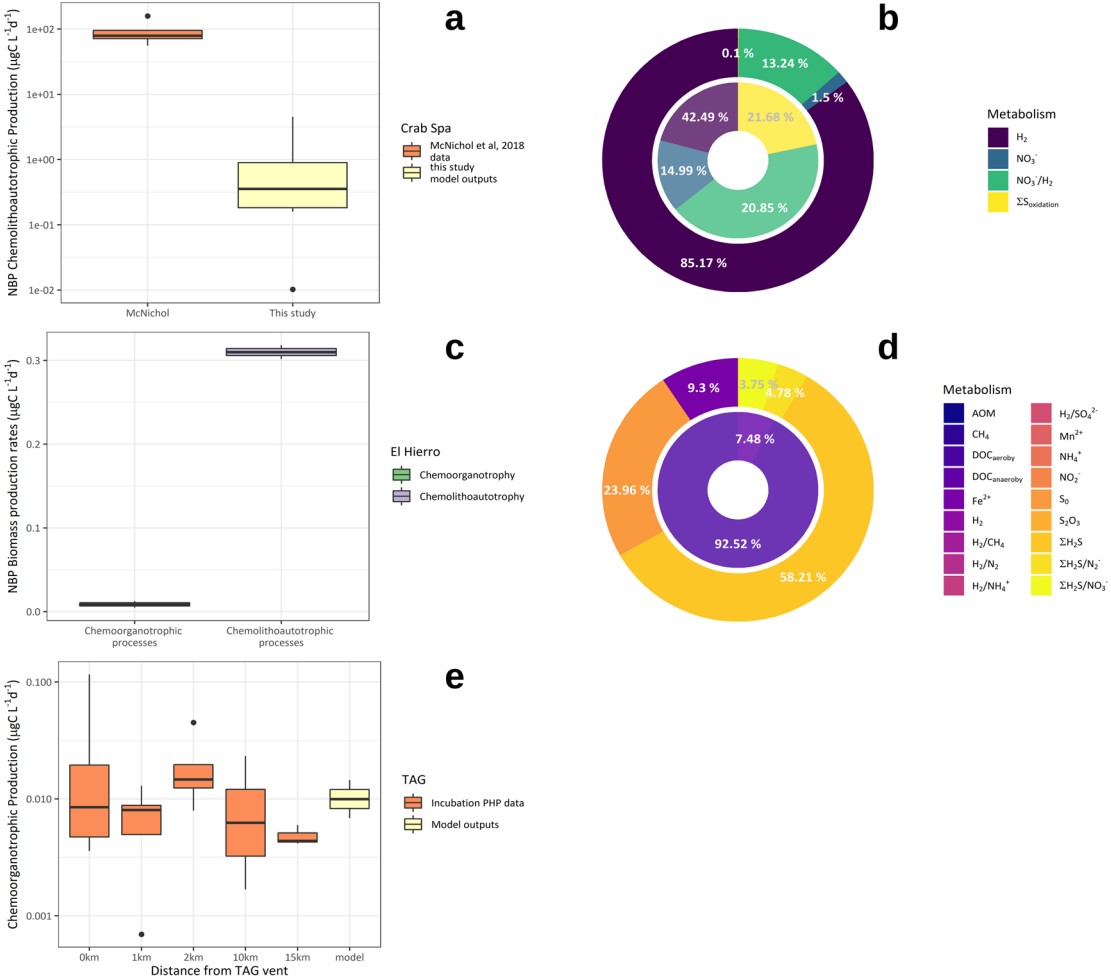

**Fig. 1 Chemolithoautotrophic and chemoorganotrophic biomass production rates in hydrothermal non-buoyant plumes: model outputs and measurements. a** Chemolithoautotrophic biomass production rates from the CrabSpa vent field: comparison between NBP model outputs and fluid incubation data from ref. [14]. **b** Distribution of the type of metabolisms responsible for the primary production at the CrabSpa vent field: comparison between NBP model outputs and fluid incubation data from ref. [14]. **c** Chemoorganotrophic and chemolithautotrophic biomass production predicted for the El Hierro eruption based on data from ref. [44]. **d** Distribution of the type of metabolisms responsible for the chemoorganotrophic and chemolithoautotrophic biomass production predicted for the El Hierro eruption based on data from ref. [44]. **e** Chemoorganotrophic biomass production rates within the TAG vent non-buoyant plume: comparison between model outputs and data derived from ${}^3$H-Leucine incorporations (conventionally designed as prokaryotic heterotrophic production, PHP) rates on incubations performed during the BICOSE 2 cruise. Boxplots denote the median (center line) and interquartile range (box), with whiskers extending to three times the interquartile range and points indicating values outside this range.

**Chemoorganotrophic production rates in hydrothermal plumes.** Our model predicts prokaryotic biomass production through chemoorganotrophic processes[18] in hydrothermal plumes higher than in surrounding seawater by a factor 2 (Table 2). Based on DOC remineralization (acetate-based-aerobic respiration and denitrification, Table 1), chemoorganotrophic production is usually designed as Prokaryotic Heterotrophic Production (PHP)[18,48]. PHP rates measured in the deep Atlantic seawater during the BICOSE 2 cruise using ${}^3$H-Leucine incorporation give rates of $4.8 \pm 1.0 \times 10^{-3}$ µgC l${}^{-1}$ d${}^{-1}$ (Fig. 1) similar to our model outputs for background conditions of $6.7 \times 10^{-3}$ µgC l${}^{-1}$ d${}^{-1}$, and to previously reported values for bathypelagic waters of the deep Atlantic water masses ($2–6 \times 10^{-3}$ µgC l${}^{-1}$ d${}^{-1}$ -[38,39,49,50]). Predicted PHP rates in the non-buoyant hydrothermal plumes from all 10 hydrothermal sites tested range from 0.0088 to 0.0124 µgC l${}^{-1}$ d${}^{-1}$ (Fig. 2), with model outputs at the TAG site up to 0.0116 µgC l${}^{-1}$ d${}^{-1}$ in close agreement with measurements performed during the BICOSE 2 cruise (Fig. 1). Elevated PHP rates in hydrothermal plumes are sustained by concentrations of labile DOC higher than surrounding seawater yields. This labile DOC in the plumes may originate from the fluids itself through $CO_2$ reduction in volatile acids (e.g., acetate, formate), abiotic formation during mixing with seawater or mobilization of buried or thermally altered organic matter, or from lateral and vertical entrainment from the diffuse vent areas where secretion by macrofauna or subseafloor or microbial activity also lead to high bulk DOC concentrations[28,51–53]. Heterotrophic microbial communities in the deep ocean have the ability to quickly respond to fresh labile C inputs[54] after long period of starvation[39], by persisting and maintaining functionality for a long time and quickly reactivating during occasional pulses of organic matter. Deep-sea marine communities found in hydrothermal plumes, such as Gamma-proteobacteria (e.g., SUP05 clade), Epsilonproteobacteria, Thaumarchaeota, or other planktonic prokaryotes[29], are able to shift between anabolic pathways and to process DOC very efficiently through mixotrophy[9,31,55–57]. Similar to vertical DOC fluxes that represent a significant part of the deep ocean respiration, we therefore propose that, by promoting inputs of labile DOC into starved deep-ocean microbial communities through direct

**Table 2 Compilation and global estimates of microbial biomass production rates through chemoorganotrophy[a] within hydrothermal non-buoyant plumes (NBP).**

Model-based microbial biomass production rates in non-buoyant plumes based on chemoorganotrophy[a] ($\mu$gC L$^{-1}$d$^{-1}$)

| Site | Rates | |
|---|---|---|
| Seawater | 0.0067 | |
| TAG | 0.0116 | |
| Rainbow | 0.0124 | |
| BrokenSpur | 0.0088 | |
| Logatchev | 0.0092 | |
| Ashadze | 0.0101 | |
| Tour Eiffel | 0.0109 | |
| EPR Grand Bonum | 0.0088 | |
| Dante | 0.0097 | |
| Edmonds | 0.0091 | |
| Kairei | 0.0092 | |
| Average ± standard error | 0.01235 ± 0.003 $\mu$gC L$^{-1}$d$^{-1}$ | |
| **Areal estimates of microbial biomass production based on chemoorganotrophy[a] (gC m$^{-2}$ y$^{-1}$)—plume height considered: 200 m** | | |
| Average ± standard error | 0.902 ± 0.219 gC m$^{-2}$ y$^{-1}$ | |
| Global estimates of microbial biomass production based on chemoorganotrophy[a] (GtC y$^{-1}$) | | |
| Number of active vents at Mid Ocean Ridges[15] | | |
| Average | Min | Max |
| 1305 | 713 | 1853 |
| **Global production rates—Plume area considered: 1000 km²** | | |
| Average | Min | Max |
| 1.2 ± 0.3 × 10$^{-3}$ GtC y$^{-1}$ | 0.6 ± 0.2 × 10$^{-3}$ GtC y$^{-1}$ | 1.7 ± 0.4 × 10$^{-3}$ GtC y$^{-1}$ |
| **Global production rates—Plume area considered: 10% of 400 × 10³ km² + 90% of 1000 km²** | | |
| Average | Min | Max |
| 0.048 ± 0.012 GtC y$^{-1}$ | 0.026 ± 0.006 GtC y$^{-1}$ | 0.068 ± 0.017 GtC y$^{-1}$ |

[a]Commonly defined as prokaryotic heterotrophic production (PHP, see main text).

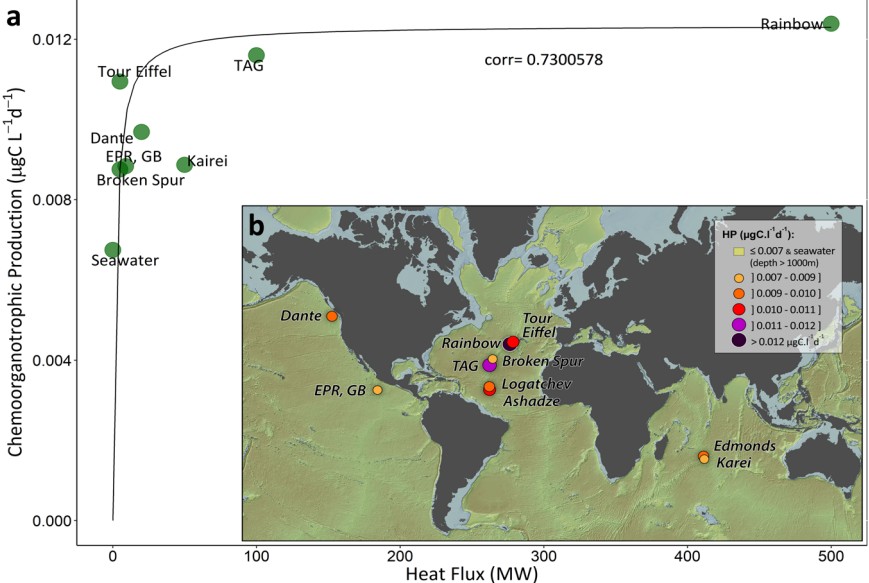

**Fig. 2 Global prokaryotic biomass production through chemoorganotrophic processes within non-buoyant hydrothermal plumes. a** Chemoorganotrophic production rates as a function of heat flux for various hydrothermal sites from the Mid-Atlantic Ridge (MAR: Rainbow, Tour Eiffel, Broken Spur, TAG), the East-Pacific Rise - Grand Bonum site (EPR, GB), the Main Endeavor vent field (Dante), and the Central Indian Ridge (CIR: Kairei). **b** Global map of the chemoorganotrophic production rates within the non-buoyant plume for the following hydrothermal sites on the MAR (Rainbow, Tour Eiffel, Logatchev, Broken Spur, TAG, Ashadze), the EPR (Grand Bonum, EPR, GB), the Main Endeavor vent Field (Dante), and the CIR (Kairei, Edmonds).

incoming or internal recycling, hydrothermal plumes increase their overall carbon processing rates.

**Global Estimates of hydrothermal prokaryotic POC production rates through chemoorganotrophy and consequences for the deep-ocean carbon standing stock.** Given the concordance

between our predicted biomass production rates and the measured values, in particular regarding the chemoorganotrophic processes (Fig. 1, Table 1), we used our model to calculate the hydrothermal plumes contribution to the dark ocean respiration. Estimated PHP rates at each hydrothermal sites were correlated with the corresponding heat fluxes (corr = 0.730, $n = 8$, Fig. 2). Using the definite integral of the empirical function we found over the global

range of hydrothermal heat flux, we provide an average value of $0.012\ \mu gC\,l^{-1}\,d^{-1}$ to be used as the first estimate for prokaryotic biomass production through chemoorganotrophy (PHP) in hydrothermal plumes over all mid-ocean ridges (Fig. 2, Table 2). Considering an average plume height of 200 m, hydrothermal prokaryotic carbon demand rates (i.e., net balance between organic matter assimilation and mineralization rates both aerobic and anaerobic) of $0.9\ gC\,m^{-2}\,y^{-1}$ (Table 2) are similar to the POC export fluxes (ca. $3\ gC\,m^{-2}\,y^{-1}$) reported in the deep Atlantic and Pacific oceans[5,58]. Our results hint that hydrothermal plumes play a critical role in the carbon balance of the dark ocean by providing local in situ POC production rates of the same order of magnitude as the vertical POC flux. By promoting the supply of labile carbon to the deep ocean, hydrothermal plumes provide microbial niches of enhanced microbial activity.

In order to extend our quantitative estimates of hydrothermal plume prokaryotic biomass production rate to a global perspective, we used an estimate of the distribution of active vents[59] to calculate global deep-ocean POC production fluxes due to chemoorganotrophic activity within hydrothermal plumes. The range of these estimates is quite broad given the uncertainties of the plume volumes to be considered. Based only on small scale plumes of about 1000 km² (i.e., ≈18 km radius if we assume a circular shape distributed over the vent site) and 200 m high, this leads to prokaryotic POC production in hydrothermal plumes ca. $0.001\ GtC\,y^{-1}$ through chemoorganotrophy which is only about 1‰ the total ocean respiration of the deep ocean[1]. However, this represents up to 10% of the POC flux estimated at 1000 m[60] and a significant part of the global deep ocean organic carbon inventory[61]. Recent studies have demonstrated export of hydrothermal plumes far beyond a few tenth of kilometers of their source, implying basin-scale transport over thousands of kilometers[6]. Such large plumes may not represent the vast majority of the distribution of hydrothermal plumes, but assuming they only account for 10% of the global hydrothermal plumes volume, it leads to $0.05\ GtC\,y^{-1}$ of plume chemoorganotrophic POC production (Table 2). These estimates represent 3% of the total carbon processed in the dark ocean through respiration and is in the same order as the vertical POC flux at 1000 m. In addition, diffuse venting, which accounts for 50–90% of the global vent flux, is not included in our calculation and might increase further the estimated rates. Additional investigations are needed to refine and better constrain our estimates, but our results highlight hydrothermal plumes as significant contributor to the overall carbon assimilation rates in the deep ocean. Whether this carbon comes from biomass produced through symbiotic chemolithoautotrophy in hydrothermal vents habitats and vertically and laterally entrained in the plume or produced in the subseafloor by the deep biosphere and thermal alteration has very different implications for the global carbon cycle. By promoting microbial activity and providing labile organic carbon in the food-limited deep-sea, hydrothermal plumes may represent a source of microbial biomass in the deep ocean that must be accounted for in global POC fluxes in order to better constrain global ocean carbon budgets.

## Methods

**Sample collection and processing at the TAG hydrothermal site**. Samples from the TAG hydrothermal vent were collected during the BICOSE1 and 2 cruises[62,63] performed in 2014 and 2018, respectively, on board the R/V Pourquoi Pas? Hydrothermal fluids were sampled during the BICOSE1 cruise[62] using 750 mL Ti syringes deployed by pairs and manipulated by the arm of the ROV Victor 6000 to collect the fluids from the vent exit. An autonomous temperature probe (NKE) was attached to the snorkel of each Ti syringe to provide temperature measurement during sampling. The Ti samplers were conditioned to avoid out-gassing and gas leakage during recovery and gas extraction on board. Immediately after recovery, pH and H₂S content were measured and total gas was extracted as described in refs. [64–66]. After gas extraction, hydrothermal solutions were transferred to acid-cleaned high-density polyethylene (HDPE) flasks for mineral composition analysis on shore (major, minor, and trace elements). Preliminary major gases ($CO_2$, $H_2$, $CH_4$, and $N_2$) concentrations were obtained on board by using a portable chromatograph (Microsensor Technology Instruments Inc.) that was mounted on line with the gas extractor. Extracted gases were conditioned on board in stainless steel pressure-tight flasks and stored until analyses. Gases were separated by Gas-Chromatography (Agilent GC 7890 A, Agilent Technologies) and quantitatively analyzed by triple detection using mass (MS 5975 C, Agilent technologies), flame ionization, and thermal conductivity detectors. Analytical precision for the $O_2$, $N_2$, $CH_4$, and $CO_2$ were 1% and 0.5% for $H_2$ (relative standard deviation based on 10 injections of a standard).

DOC measurements were performed using a multi N/C 3100 from Analytic Jena®. Analytical precision was 1% (relative standard deviation on 30 repetition of a standard measurement).

Samples from the non-buoyant plume at TAG for heterotrophic activity rates were collected during the BICOSE2 cruise[63] using High Pressure Sampler Unit fitted on a CTD carrousel[67].

**Total cell counts**. Samples for total cell direct counting were fixed and homogenized in a 2% (v/v) formaldehyde saline solution (1 ml of seawater in 9 mL of solution) on board the ship. As described by ref. [68], a 0.5–3 ml aliquot was stained using 50 µL 2X SYBR Gold (Invitrogen) in 10 mL of sterile 2% formaldehyde saline solution for 3 min, and then filtered onto a black polycarbonate membrane filter (0.22 µm pore size). Cells were counted under an epifluorescence microscope (Zeiss AxioImager.Z2 microscope – Göttingen, Germany).

**Heterotrophic activities and cell densities abundance**. Prokaryotic heterotrophic production (PHP), similar to chemoorganotrophic microbial biomass production, was measured by incorporating L-[4,5-³H]-Leucine (³H-Leu, 109 Ci mmol⁻¹ of specific activity, PerkinElmer®) to get a final concentration of 10 nM in high-pressure bottles as described in ref. [67]. To calculate the PHP, we used the empirical conversion factor of 1.55 ng C pmol⁻¹ of incorporated ³H-Leu according to Calvo-Diaz and Moran[69].

**Hydrodynamic model**. Our hydrodynamic modeling is a simple 1-D buoyant plume model based on the Morton and Turner definition under the Boussinesq approximation: past the first meters of advection driven by the initial fluid velocity, the main driving force of the rising plume is the density difference between the hydrothermal fluid and the surrounding seawater[26,27]. Such models have been used widely in the literature[70,71].

Briefly, we consider a turbulent plume formed by the steady release of a buoyant isothermal fluid into a quiescent environment. The environment is assumed to be unconfined and stratified. Ambient seawater at the vent site field depth is of uniform density $\rho_{sw\ site}$. Reference seawater density for stratification frequency calculations ($\rho_{sw\ ref}$) is taken above the non-buoyant plume height. Through the lateral entrainment process of external fluid across the plume boundary ($u_e$), the mass flow rate increases with height ($z$). At the same time, plume radius $r$, vertical velocity $u$, and density $\rho$ may evolve with $z$.

The governing equations for mass, momentum, and density deficit conservation for a steady plume rising in a stable uniform environment in terms of its characteristic radius $r_z$, vertical velocity $u_z$, and density $\rho_z$ can be written in the following forms:

$$\frac{d}{dz}\left[u_z r_z^2\right] = 2r u_e \quad (1)$$

$$\frac{d}{dz}\left[u_z^2 r_z^2\right] = \frac{\rho_0 - \rho_z}{\rho_0} g r_z^2 \quad (2)$$

$$\frac{d}{dz}\left[(\rho_0 - \rho_z)\, u_z r_z^2\right] = 0 \quad (3)$$

Lateral entrainment $u_e$, proportional to the vertical water velocity is defined with $\alpha_e$ the lateral entrainment coefficient as follows:

$$u_e = \alpha_e . u_z \quad (4)$$

$\alpha_e$ value is mostly included between 0.05 and 0.2, and may vary between two hydrothermal plumes or even within the same plume. For a pure jet, where mixing is driven by initial advection, $\alpha$ (designed by $\alpha_j$) has low values ranging between 0.05 and 0.10. A pure buoyant plume (i.e., mixing is driven by density anomaly), $\alpha$ (designed by $\alpha_p$) should be higher in the initial phase (0.15–0.20) and then decrease after the buoyancy initial acceleration phase[71]. Hydrothermal plume water entrainment ranges between these two behavior schemes, and balance between pure jet and pure buoyant plume behavior is done by calculating the Froude number (Fr) which represent the ratio of the vertical velocity by the buoyancy force $g'$ (Eqs. (5) and (6)), where $g$ represents the universal gravitational constant.

$$g' = g \cdot \frac{\rho_z - \rho_0}{\rho_0} \quad (5)$$

$$Fr = \frac{u_z}{\sqrt{(g' \cdot R)}} \quad (6)$$

The following semi-empirical Eq. (7) is used to predict the evolution of the entrainment coefficient $\alpha_e$ of the rising plume[71]:

$$\alpha_e = \alpha_j - \left(\alpha_j - \alpha_p\right) \cdot \left(\frac{\mathrm{Fr}_p}{\mathrm{Fr}}\right)^2 \tag{7}$$

With $\mathrm{Fr}_p$, the Froude number for a pure buoyant plume.

This advection-only transport model is defined on a 1D volumetric grid that allows resolving temperature, salinity, and dilution ratios at each grid cell. Number of cells is arbitrarily fixed to 300. At each site, input parameters of the model are fluid temperature and salinity, diameter of the vent, and fluid velocity at the output, which will drive the initial buoyancy flux. These data were collected from the literature (see Table SI.1), along with the density values needed for stratification frequency calculations[72-77]. Hydrothermal sites were chosen so that both physical (salinity, temperature, vent diameters, and velocity output) and chemical data (substrate concentrations see Table SI.1) of the vents were available.

**Thermodynamic microbial model based on MTS theory[16,17].** Our growth microbial model is strictly based on the formalism of the MTS theory of growth derived from first principles of thermodynamics of growth[16,17]. A detailed presentation of the framework and governing equations is provided in ref.[16] and we encourage the reader to refer to this publication for a thorough description of the model, as we will only describe its main components and governing equations.

Here, we model the non-buoyant plume microbial community. Plume height and chemical species concentration are based on dilution with seawater based on our hydrodynamic outputs: when hydrodynamic outputs are not available, an arbitrary $10^5$ dilution factor with seawater is applied[28]. In total, we consider 21 chemical species in our model (see list in Table SI.1).

The microbial community is subdivided into 18 guilds, each corresponding by the metabolism it catalyzes (see Table 1). Population density of each guild is represented by the molar concentration of the generic C-normalized biomass molecule $C_1H_{1.8}O_{0.5}N_{0.2}$.

The model stoichiometry is formulated using a vectorial approach. Let $r$ be the number of reagents involved in the system: in our system, $r$ equals 39 (i.e., 21 chemical species and 18 microbial guild biomasses). Let $C$ be a $r \times 1$ vector storing the concentration of all reagents of the system, including biomasses, in mol.m$^{-3}$ at a given time (s).

Let $p$ be the number of processes affecting the concentrations of the reagents; the derivative of $C$ over time is expressed from the balance equation of $C$ as

$$C = A_{met}*R_{met} + A_{death}*R_{death}$$

where $A_{met}$ and $A_{death}$ are $r \times p$ matrices storing the (unitless) stoichiometric coefficients of, respectively, metabolic and mortality process for every reagent, $R_{met}$ and $R_{death}$ are the $p \times 1$ vectors of the rate (in s$^{-1}$) of every metabolic and death processes and $*$ denotes the matrix product operator. By convention, the stoichiometric coefficients are either positive or negative depending on the production or consumption of the corresponding chemical species, respectively.

$A_{met}$ is therefore the matrix of dimension $39 \times 18$ storing the stoichiometric coefficients of the metabolism of every guild, and $R_{met}$ the $18 \times 1$ vector of the rate of each guild reaction, whereas $A_{death}$ is the matrix of dimension $39 \times 18$ storing the stoichiometric coefficients of mortality of every guild, and $R_{death}$ the $18 \times 1$ vector of the mortality rates $\alpha$ of each guild ($\alpha$ taken equal for all guilds). Mortality is defined by a first-order term for microbial biomass[15] and we assume that for 1molCBiomass released, 40% is recycled into labile DOC (DOC$_l$), 10% into refractory DOC (DOC$_r$), and 50% goes into the POC pool.

$A_{met}$ is a linear combination of two matrices $A_{ana}$ and $A_{cat}$, both of dimensions $r \times 18$, respectively, storing the coefficients of the anabolic and catabolic reactions, and adjusted to close the elemental balance in each reaction separately. Following Delattre et al.[16], the stoichiometric coefficients of a catabolic reaction are so that exactly one electron donor molecule is consumed. The stoichiometric coefficients of an anabolic reaction are so that exactly one biomass molecule is produced.

The Gibbs free energy of formation of every chemical species used in the simulations is calculated using the thermodynamic database from the CHNOZ package and the temperature and pressure conditions issued from the hydrodynamic model.

The overall metabolism of the whole microbial community is expressed as followed (Eq. (8)):

$$A_{met} = A_{ana} + \lambda A_{cat} \tag{8}$$

where $\lambda$ is the number of times the catabolic reaction of a guild has to be performed for the total produced energy to meet the energy barrier of growth requirement (expressed as mol.DonormolBiomass$^{-1}$). $\lambda$ is a diagonal matrix of guild-specific scalar factors (denoted $\lambda_g$) that ensures the balance between stoichiometry and energy reactions. $\lambda_g$ definition follows Kleerebezem's formalism summarized in Eq. (9)[78]:

$$\lambda_g = -\frac{\Delta G_{ana} + \Delta G_{diss}}{\Delta G_{cat}} \tag{9}$$

where $\Delta G_{ana}$ is the Gibbs free energy change for the anabolic reaction, $\Delta G_{cat}$ is the Gibbs free energy change for the catabolic reaction and $\Delta G_{dis}$ is the dissipated free energy of growth. Only exergonic catabolic reactions can lead to growth in our

model: therefore $\Delta G_{cat}$ is set to zero if it happens to be positive during calculation. $\lambda_g$ factors are computed at each time step of system integration. Water and biomass activities are included in the mass and action ratio calculated following Helgelson's definition[30]. $\Delta G_{ana}$, $\Delta G_{cat}$, and $\Delta G_{diss}$ values are expressed in kJ.C-molBiomass$^{-1}$.

The dissipated free energy is the Gibbs free energy assumed to be identifiable with the variable $-Y_{GX}^{max}$ as empirically defined by Heijnen[47] as

$$\Delta G_{diss} \approx -Y_{GX}^{max}$$
$$\Delta G_{diss} = 200 + 18(6-2)^{1.8} + \exp\{[(3.8-8)^2]^{0.16} * (3.6 + 0.4 \times 2)\} \tag{10}$$

Where 2 and 8 are, respectively, the length chain and degree of reduction of acetate, the carbon source for heterotrophic growth considered in our model[16,47]. This leads to $\Delta G_{diss}$ values of 1477 kJ molCBiomass$^{-1}$. Given the range of variations for $\Delta G_{diss}$ depending on the type of metabolism consider, we chose to use the empirical values from the original publication for heterotrophic metabolisms ($\Delta G_{diss} = 539-557$ kJ molCBiomas$^{-1}$) and chemolithoautotrophic metabolic pathways including reversed electron transport (average $\Delta G_{diss}$ considered: 3500 kJ molCBiomass$^{-1}$).

The growth rate function used in the simulations is the multi-substrate growth rate function[16,17] that arises from several simple principles regarding microbial growth:

– before achieving cell division, microbs must reach a fixed energy threshold that can be broken down into anabolic energy $\Delta G_{ana}$ and dissipated energy $\Delta G_{diss}$.
– the energy available to overcome this energy threshold (or barrier) is the catabolic energy $\Delta G_{cat}$ obtained from the catabolism of substrate molecules
– substrate molecules are considered as particles randomly distributed around the cells
– if a fixed fictional and so-called harvest volume around the cell ($V_{harv}$) contains enough substrate to reach the energy threshold, the cell is said to be in an "activated" state and only activated cells are able to divide.

Given the above assumptions, the proportion of activated cells at a given time is expressed using a probabilistic approach[16,17]. For a given guild, the formula of the microbial growth rate $\mu$ (s$^{-1}$) is defined as

$$\mu = \mu_{max} \prod_i e^{\frac{A_{met,i}}{V_{harv}[S_i]}} \tag{11}$$

where $A_{met,i}$ is the negative stoichiometric coefficient of substrate $i$ (mol.C-mol-Biomass$^{-1}$) computed in Eq. (8), and $[S_i]$ the concentration of substrate $i$ (mol.m$^{-3}$). The value of $\mu_{max}$ for every guild was set to $\frac{k_B T}{h_P}$ where $k_B$ is the Boltzmann constant (Table SI.2), $T$ is the temperature of the system and $h_P$ the Planck constant[16].

The $18 \times 1$ $R_{met}$ vector storing the rate of each metabolic reaction is

$$R_{met} = \text{diag}(M)*[X] \tag{12}$$

where $M$ is the $18 \times 1$ vector of the microbial growth rate of each guild considered and $[X]$ the $18 \times 1$ vector of the biomass concentration of each guild.

Similarly, the $18 \times 1$ $R_{death}$ vector storing the rate of each cell death is

$$R_{death} = \text{diag}(M_{death})*[X] \tag{13}$$

with $M_{death}$ is the $18 \times 1$ vector of the microbial mortality rate $\alpha$ of each guild. This ordinary differential equation system is implemented and solved using R and the ReacTran and CHNOZ packages[79-81].

**Sensitivity analysis.** We performed sensitivity analysis for our microbial model to test the influence of the parameters on our model outputs. We tested the effect of temperature, salinity, pressure, $V_{harv}$, and Mortality rate $\alpha$ over the following range for the Crab Spa vent field initial conditions: 2–50 °C, 15–45‰, 100–500 bars, 1000–10,000 m$^3$ molBiomass$^{-1}$ and $0.1 \times 10^{-8}$–$1 \times 10^{-6}$ s$^{-1}$). Except for $V_{harv}$, parameters did not show a large influence on our model outputs, with variation of biomass production rates within 20% for all metabolic pathways, except for H$_2$ and NH$_4$ metabolisms that showed variations >50%. $V_{harv}$, however, did influence greatly the overall biomass production rates predicted by the model: Figure SI.2 shows the variation of the 18 different microbial metabolisms considered over the span of 300 h with $V_{harv}$ ranging from 1000 to 10,000 m$^3$ molBiomass$^{-1}$. We therefore fixed it to 1000 m$^3$ molBiomass$^{-1}$ for all our calculations.

**Chemolithoautotrophy metabolism: 300 days runs.** In order to look further into microbial processes and chemolithoautrophy in hydrothermal plumes, the model was ran for 300 days at the TAG and Endeavor Main Field sites. Endeavor Main Field plume substrate concentrations (in particular NH$_4$$^+$ and CH$_4$) were taken from the literature[41,42]. Although no net primary production (i.e., net DIC incorporation) was observed, positive chemoautotrophic rates were found (Fig. SI3). At the Endeavor Main Field Sites, methane oxidation rates were up to 1.3 nM d$^{-1}$. Although non-null, the model predict only very little chemoautotrophic activity in the TAG hydrothermal non-buoyant plume.

**Global extrapolation of heterotrophic production in hydrothermal plumes.** The function fitted between heterotrophic production (HP) rates (µgC L$^{-1}$ d$^{-1}$) and

*Heat Flux* (MW) was the following:

$$HP = 0.0123 \frac{\text{Heat Flux}}{5.987 + \text{Heat Flux}} \quad (14)$$

Residual standard error was 0.003 and correlation 0.77 ($n = 8$, $p < 0.11$). The average *HP* rates for hydrothermal vents considered for global extrapolation was therefore calculated using the global estimates for hydrothermal vent heat flux ($\approx 10^{12}$ W)[82], a value of 0.001 W for surrounding seawater *Heat Flux* and the primitive of the function defined in Eq. (14).

$$\text{Average}_{HP} = \frac{1}{Q_{vents} - Q_{sw}} \int_{Q_{sw}}^{Q_{vents}} HP(Q)dQ \quad (15)$$

With

$$\int_{Q_{sw}}^{Q_{vents}} HP(Q)dQ = [0.0123*Q - 0.0123*5.987* \ln(Q + 5.987)]_{Q_{sw}}^{Q_{vents}} \quad (16)$$

**Assessing the error of our modeling approach on the distribution of substrates within the NBP.** We compared the concentrations of substrates predicted by our hydrodynamic model within the NBP with data available in the literature. When predicted concentrations were out of the available data range, we performed simulations with updated substrate concentrations to quantify the impact on our predicted biomass production rates. All NBP substrates were kept identical except for chemical species available in the literature whose concentrations were taken as reported. The species concerned are methane, hydrogen, dissolved iron, dissolved manganese, total hydrogen sulfide, ammonia, nitrate, and nitrite. Results are compiled in Supplementary Table 3. Predicted NBP substrate concentrations were out of range for the Rainbow, Broken Spur, and Logatchev hydrothermal vent sites, with no impact on heterotrophic biomass production rates (<0.5% rates difference between updated and non-updated substrate concentrations). The only significant difference occur for chemoautotrophic production rates at the Rainbow vent sites with predicted rates of 0.0074 µgC l$^{-1}$ d$^{-1}$, for not updated substrates concentrations (v.s. 0.0000 for updated ones). Such difference is due to the much larger concentrations of substrates. For the Rainbow NBP, we therefore considered the chemoautotrophic production rates obtained from updated concentrations.

## Data availability

All data generated or analyzed during this study are included in this article and its supplementary information files. Source data for Fig. 1a, 1b, 1c and 1d are available as datasets or model outputs using the NBPmicrob R package available online (https://doi.org/10.5281/zenodo.4587935). Samples list from the BICOSE1 and 2 cruises are available online. Source data are provided with this paper.

## Code availability

The NBPmicrob R package corresponding to the model used is deposited on Zenodo (https://doi.org/10.5281/zenodo.4587935).

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

## Acknowledgements

We thank C. Brandily and S. Fuchs for providing assistance and help during the BICOSE 1 (https://doi.org/10.17600/14000100) and 2 (https://doi.org/10.17600/18000004) cruises. We thank A.-S. Alix for her help in creating Fig. 2b. We thank the chief scientist M.-A. Cambon and all scientific parties, the crew of R/V Pourquoi Pas? and Victor6000 ROV and Nautile crew for their work and support during the BICOSE 1 and 2 cruises. We also thank E. Desmond-Le Quemener for his advice and discussion regarding the microbial model. This work was partially supported by the European Union Seventh Framework Programme (FP7/2007–2013) project Managing Impacts of Deep-seA reSource exploitation (MIDAS), grant agreement 603418 and by the internal Ifremer projects REMIMA and MERLIN-ABYSS. This work was also supported by the "Laboratoire d'Excellence" LabexMER (ANR-10-LABX-19) and co-funded by a grant from the French government under the program "Investissements d'Avenir". Post-doctoral fellowship grant by the FP7-MIDAS project and the Scientific Department of Ifremer.

## Author contributions

C.C. prepared the manuscript, performed modeling simulation (writing and running the model) and global extrapolation, and planned part of the project. A.P. contributed to the modeling effort (writing of the hydrodynamic model). J.G. and G.R. provided expertise on hydrodynamic modeling. V.G. and J.-P.D. performed gaz measurements on hydrothermal fluids from the BICOSE 1 and 2 cruises. V.C., E.R., A.G., C.T., and M.G. collected and provided prokaryotic heterotrophic production, microbial abundances data, and expertise in microbial metabolism. P.-M.S. was in charge of planning part of the project and mentoring.

## Competing interests

The authors declare no competing interests.

**Additional information**

