## [Peer Review File · Nature Communications]

Hydrothermal plumes as hotspots for deep-ocean heterotrophic microbial biomass productionREVIEWER COMMENTS

Reviewer #2 (Remarks to the Author):

The presented manuscript sheds new light on how carbon is being produced, consumed, and cycled in hydrothermal systems. This is an important question within the hydrothermal field and in our understanding of the biogeochemical impact hydrothermal systems have on global geochemical cycles. This manuscript represents an important contribution to the literature. With that in mind, there are a few areas that need addressing. I have provided both general comments and line edits below.

1. Please carefully proofread and address grammar in the manuscript.
2. Please either use non-buoyant or neutrally buoyant through the manuscript and supplemental information.

Abstract is great, clear and to the point.

Line 45: dissolved or particulate organic carbon?

Introduction

I would recommend using transition words sparingly. As a reader, it became a little difficult to follow the main points/take-home messages within the introduction.

Line 92: Capitalize the e after 1.16 or change to $\times 10^{-8}$

Section-Chemoautotrophic rates in hydrothermal plumes

Sentence structure and grammar need to be revisited in this section. Please be careful of run-on sentences.

Line 149: add 'our' after 'column', add 's' to 'predicts', and remove 'only'

Line 150: do you mean 'which is consistent with in-situ observations'?

Line 150-153: This sentence is confusing. Please rework.

Line 153: add 'our' before 'model'

Line 176: remove 'therefore'

Section-Heterotrophic rates in hydrothermal plumes

Line 181-182: replace 'up to' with 'by'

Figure 2: Please label the vent sites on the insert or make specific sites specific colors. Also, do the colors of the circles on the insert have a specific meaning?

Line 195: Include citation Bennett et al. 2011-Dissolved and particulate organic carbon in hydrothermal plumes from the East Pacific Rise, 9°50'N, which also discusses sources/sinks of DOC within hydrothermal plumes.

Line 195-196: Please rework the sentence beginning with 'Starved microbial communities...'. It is confusing in its current form.

Line 197: add 'in' between found and hydrothermal

Line 199: What is the difference between universal mixotrophy and mixotrophy?

Line 201: did you mean deep ocean respiration?

Line 201: remove "that"

Section-Global Estimates of hydrothermal respiration rates and consequences for the deep-ocean carbon standing stock

Line 229: add "a" or 'the' before hydrothermal plume

Line 248: into not in

Supplemental Information

Line 7: is there supposed to be a question mark after Pas?

Line 45: literature not litterature

Line 90: Please write out the number 21 and this sentence is a fragment. Please rephrase.

Line 97: put paratheses around "i.e. 21 chemical...guild biomass"

Reviewer #3 (Remarks to the Author):

Cathalot et al. describe the challenge of determining the biogeochemical interactions between hydrothermal systems and the deep ocean. They propose a model that focuses on the role of hydrothermal plumes in carbon cycling and compare to known vent studies. Their model provides the first estimate for heterotrophic biomass production in hydrothermal plumes over all MOR. They address a community need for better understanding how crustal fluids interact with the ocean, which a specific focus on the "plume interface" between the two.

Main issues:

It is unclear throughout when the paper is describing anabolism or catabolism, and which model parameters are selected based on anabolic rates vs catabolic rates. For example, analyses of ^{13}C -POC from a ^{13}C -DIC incubation would be anabolism and does not require autotrophic organisms to generate the label (See Erb et al Carboxylases in Natural and Synthetic Microbial Pathways). Similarly, there are many vent "autotrophs" that can also incorporate organic carbon. Therefore, the manuscript would be better served to strictly describe DOC and DIC incorporation (anabolism) or respiration and oxidation (catabolism) rather than "autotrophy" and "heterotrophy."

The study would be improved by further integration and comparison with relevant vent literature.

Comparison to fluxes from cool crustal fluids:

Shah et al. "Microbial decomposition of marine dissolved organic matter in cool oceanic crust"

Other plume papers:

Lin et al. "Intense but variable autotrophic activity in a rapidly flushed shallow-water hydrothermal plume (Kueishantao Islet, Taiwan)"

Sheik et al. "Spatially resolved sampling reveals dynamic microbial communities in rising hydrothermal plumes across a back-arc basin"

Organic matter characterization papers:

Rossel et al. "Bioavailability and molecular composition of dissolved organic matter from a diffuse hydrothermal system"

These vent papers include SIP with organic substrates and do have plume data (e.g. metatranscriptomes) that could inform in situ activity, though SIP was not published for the plumes:

- Winkle et al. "Identification and activity of acetate-assimilating bacteria in diffuse fluids venting from two deep-sea hydrothermal systems"
- Trembath-Reichert et al. "Active seafloor microbial communities from Mariana back-arc venting fluids share metabolic strategies across different thermal niches and taxa"

Calculations of precision/accuracy/error missing from geochemical measurements in supplement.

Paragraph starting at Ln 90 has many estimates without any numbers provided. What would be the numerical range for "slightly overestimated." It was also not clear why only chemoautotrophic rates

may be overestimated. Potential error ranges on both DOC and DIC rates would aid in this interpretation rather than qualitative statements.

Minor: Abstract should include rates in volume per year for those that think in those terms as opposed to just focusing on fluxes.

There are spelling errors throughout the supplement and minor grammatical and formatting errors in the main manuscript.

Ln 92: $1.16e-8 \text{ s}^{-1}$ – what do these units mean and how was this value chosen?

Ln 175: The modeling approach provides a “lower ground” for biomass production, but this result is unclear given how few experimental data were compared and the metabolic complexity explained just previously to this conclusion. Further work to ensure this lower bound conclusion is needed.

Ln 206: “good concordance” what is meant by this? Is this being used colloquially or in the mathematical sense?

Table SI.2 1000 m³ molBiomass⁻¹ Seems very high for anything to do with a single-cell. How was this selected?

Reviewer #4 (Remarks to the Author):

The manuscript entitled “Hydrothermal plumes as hotspots for deep-ocean heterotrophic carbon fixation” presents a model for heterotrophic productivity in globally distributed hydrothermal plumes and highlights the potential importance of plume related microbial activity to global geochemical cycling, specifically that of carbon. This is a topic of significant interest to the hydrothermal vent community and those examining oceanic carbon budget.

The authors generally do a good job of addressing the limitations and caveats of their model which is appreciated. However, I would like to have seen more of this under the section “Heterotrophic rates in hydrothermal plumes”, although I do appreciate that the authors were able to frame their model results with measured data.

Overall, I believe the manuscripts merits consideration for publication.

Specific comments:

94-101: This section on the manuscript discusses that the authors do not take into account abiotic processes occurring in the buoyant plume. While I understand why taking into account these complexities is beyond the scope of the model presented, I feel these statements downplay the significant chemical alterations that occur in the nascent buoyant plume and its implications for the authors model. For example see Findlay et al. Nat Commun (2019). <https://doi.org/10.1038/s41467-019-09580-5> and Gartman et al. Nat. Geosci. (2020). <https://doi.org/10.1038/s41561-020-0579-0>. This is an important aspect of the manuscript as much of the chemistry used in this model was likely collected in the buoyant plume in close proximity to the vent opening.

110-111: Please provide references for the literature referenced here. Is this referring to Table 1?

Regarding Table 1: Please provide references for in table citations (e.g. McNichol et al 2016; Lin et al 2006).

Line 153-155: Is (13) the correct reference for this work?

Supplemental Table 1: How were labile and refractory DOC determined? Are these measured parameters or based on the estimates detailed on line 113? Also, please provide references for Crab Spa and Seawater data.

Response to reviewer #2

Figure 2 caption (Lines 436-438): Neutrally buoyant plume has been changed to non-buoyant plume.

Table 2 (Line 450): Neutrally buoyant plume has been changed to non-buoyant plume.

Line 45. Our study shows that hydrothermal plumes, by promoting heterotrophic biomass production, consume dissolved organic carbon (= sinks) and are net producers of particulate organic carbon (= source). Since we mainly focus on biomass production, we chose to put forward “sources of particulate organic carbon” (l.45-46).

Line 66. However has been removed

Line 92. $1.16e^{-8}$ has been changed to $1.16 \cdot 10^{-8}$.

Line 149: ‘our’ has been added after ‘column’, ‘s’ has been added to ‘predicts’, and ‘only’ has been removed.

Line 150: ‘consistently with in situ observations’ has been replaced by ‘which is consistent with in-situ observations’.

Line 150-153: Following reviewer 2’s comment, we reworked the sentence. We now state: “At the Endeavour site, we ran our model for 300 days to increase chemolithoautotrophic biomass production due to internal recycling and microbial turnover. However, model outputs predict methane oxidation rates up to $1.3 \text{ nmol m}^{-3} \text{ d}^{-1}$ but no net chemolithoautotrophic biomass production in the Endeavour hydrothermal plume (Fig. SI 3). ”.

Line 154: We added ‘our’ before model.

Line 177: we removed ‘therefore’.

Line 182: we added “by” as requested by reviewer 2.

Figure 2: we reworked Figure 2 according to reviewer 2’s suggestion i.e. adding labels to the vent sites on the insert. While reworking the figure, we followed the Nature Communications formatting instructions and we labeled both panels (a and b). We also assigned a detailed legend for both panels.

Line 196: Following reviewer 2’s comment, we reworked the sentence. We now state: “Heterotrophic microbial communities in the deep ocean have the ability to quickly respond to fresh labile C inputs after long period of starvation³⁸, by persisting and maintaining functionality for a long time and quickly reactivating during occasional pulses of organic matter.”

Line 199: What is the difference between universal mixotrophy and mixotrophy? Actually the complete sentence stated: “Deep-sea marine communities found in hydrothermal plumes, such as gammaproteobacteria, Thaumarchaeota or other planktonic archaea, are able to shift between metabolic pathways and to process DOC very efficiently via universal mixotrophy [all the phylum concerned or all the community] or by mixotrophy or heterotrophy in a subgroup of the phylum [only part of the phylum]”. Given reviewer’s 2 comment, we feel that this sentence was a bit confusing and reworked it. We simplified it and now state: “Deep-sea marine communities found in hydrothermal plumes, such as gammaproteobacteria, Thaumarchaeota or other planktonic archaea, are able to shift between metabolic pathways and to process DOC very efficiently through mixotrophy”

Line 201: We changed dark ocean respiration to deep ocean respiration.

Line 201: we do not see where to remove “that” without changing the meaning of the sentence.

Line 230: we added an “s” to hydrothermal plume (instead of adding “a” or “the” before hydrothermal plume as suggested by reviewer 2.

SI. Line 7: Yes there is a question mark after Pas. The name of the research vessel is “Pourquoi Pas?” which means “Why not?”.

SI. Line 45: We corrected “Literature “

SI. Line 90: Following reviewer 2’s comment, we reworked the sentence and now state: “In total, we consider 21 chemical species in our model (see list in Table SI1).”

SI. Line 98: Following reviewer 2’s comment, we put brackets around “i.e. 21 chemical... guild biomass”.

Response to reviewer #3

Reviewer#3's comment : « It is unclear throughout when the paper is describing anabolism or catabolism, and which model parameters are selected based on anabolic rates vs catabolic rates. For example, analyses of ¹³C-POC from a ¹³C-DIC incubation would be anabolism and does not require autotrophic organisms to generate the label (See Erb et al Carboxylases in Natural and Synthetic Microbial Pathways). Similarly, there are many vent "autotrophs" that can also incorporate organic carbon. Therefore, the manuscript would be better served to strictly describe DOC and DIC incorporation (anabolism) or respiration and oxidation (catabolism) rather than "autotrophy" and "heterotrophy." »

As pointed out by reviewer#3, the previous version of the manuscript lacked precision in the terms used to describe catabolism, anabolism, microbial biomass production and subsequent POC production. We reworked the different sections of the manuscript and carefully assess the metabolic and carbon processing terms we used to ensure clarity for the reader.

We now follow the terminology proposed by Bo Jorgensen in Marine Geochemistry (2nd rev., updated and extended ed. 2006 Edition by Horst D. Schulz, Matthias Zabel - SPRINGER), and use the following terms:

- « Chemolithotrophic » to refer to catabolic processes using inorganic compounds (e.g. DIC) as energy source.
- « Chemoorganotrophic » to refer to catabolic processes using organic compounds (e.g. DOC, acetate) as energy source and anabolic processes using organic compounds (e.g. DOC, acetate) as carbon source (assimilation).
- « Autotrophic » to refer to anabolic processes using inorganic compounds (e.g. DIC) as carbon source (assimilation)
- « Heterotrophic » to refer to anabolic processes using organic compounds (e.g. DOC, acetate) as carbon source (assimilation)
- « Chemolithoautotrophic » to refer to catabolic and anabolic processes using inorganic compounds (e.g., DIC)
- « Chemiolithoheterotrophic » to refer to catabolic processes using inorganic compounds (e.g. DIC) and anabolic processes using organic compounds (e.g., DOC)

We changed chemoautotroph to chemolithoautotroph throughout the revised manuscript, we also added the anabolic processes considered in the model in Table1. In the last section, we now refer to chemoorganotrophic production, and chemoorganotrophic POC or biomass production to indicate

precisely what the model outputs are, i.e. net biomass production after catabolism and anabolism (L. 216-261).

Reviewer#3's comment : « The study would be improved by further integration and comparison with relevant vent literature.

Comparison to fluxes from cool crustal fluids:

Shah et al. "Microbial decomposition of marine dissolved organic matter in cool oceanic crust"

Other plume papers:

Lin et al. "Intense but variable autotrophic activity in a rapidly flushed shallow-water hydrothermal plume (Kueishantao Islet, Taiwan)"

Sheik et al. "Spatially resolved sampling reveals dynamic microbial communities in rising hydrothermal plumes across a back-arc basin"

Organic matter characterization papers:

Rossel et al. "Bioavailability and molecular composition of dissolved organic matter from a diffuse hydrothermal system" »

Lines-147-150. Following reviewer #3's suggestion, we cited Lin et al, 2021 (ref: 45) and modified our sentence by stating that no DIC assimilation rates data were available deep hydrothermal plumes. The manuscript now reads: "Although there are no *in situ* data of chemolithoautotrophic production rates (defined hereafter as Dissolved Inorganic Carbon (DIC) assimilation rates) available in deep hydrothermal plumes to our knowledge, significant ammonia and methane oxidation rates have been reported along the Juan de Fuca ridge⁴⁰⁻⁴², and variable chemoautotrophic rates have been observed in a shallow water hydrothermal plume offshore Taiwan⁴³"

Lines 196 – 207. Following reviewer #3's suggestion, and along with reviewer #2 comment (see above), we reworked this part and added the following literature to discuss NBP microbial community, DOC lability, and microbial ability to process it (Ref 55: Shah et al, 2017; Ref 30: Sheik et al, 2015; Ref 53: Rossel et al, 2017). We added a more thorough description of the microbial groups usually found in hydrothermal plumes (e.g. citing SUP05 clade, Epsilonproteobacteria). We now state: "This labile DOC in the plumes may originate from the fluids itself through CO₂ reduction in volatile acids (e.g. acetate, formate), abiotic formation during mixing with seawater or mobilization of buried or thermally altered organic matter, or from lateral and vertical entrainment from the diffuse vent areas where secretion by macrofauna or seafloor or microbial activity also lead to high bulk DOC concentrations^{29,51-53}. Heterotrophic microbial communities in the deep ocean have the ability to quickly respond to fresh labile C inputs after long period of starvation⁴¹, by persisting and maintaining functionality for a long time and quickly reactivating during occasional pulses of

organic matter. Deep-sea marine communities found in hydrothermal plumes, such as Gammaproteobacteria (e.g. SUP05 clade), epsilonproteobacteria, Thaumarchaeota or other planktonic archaea³⁰, are able to shift between anabolic pathways and to process DOC very efficiently through mixotrophy^{13,32,54-56}.”

Calculations of precision/accuracy/error missing from geochemical measurements in supplement.

Precision of all geochemical analytical procedures were added. We now state: “Analytical precision for the O₂, N₂, CH₄ and CO₂ were 1% and 0.5% for H₂ (relative standard deviation based on 10 injections of a standard). DOC measurements were performed using a multi N/C 3100 from Analytic Jena®. Analytical precision was 1% (relative standard deviation on 30 repetition of a standard measurement).”

“Paragraph starting at Ln 90 has many estimates without any numbers provided. What would be the numerical range for “slightly overestimated.” It was also not clear why only chemoautotrophic rates may be overestimated. Potential error ranges on both DOC and DIC rates would aid in this interpretation rather than qualitative statements.”

According to reviewer#3’s and following reviewer#4’s comments (see response to reviewer 4), we added a dedicated section in the supplementary material (l. 212 – 226, Supplementary Material) to assess potential errors of biomass production rates due to our simplified hydrodynamic geochemical model with no chemical reactions considered within the rising plume. For hydrothermal vent sites where NBP substrate concentrations were available, we compared predicted values of NBP substrate concentrations from our model with data from literature. We also ran biomass production simulations for our predicted NBP values and the corresponding NBP data. Results are compiled in Table S13. No significant differences on heterotrophic nor chemoautotrophic biomass production rates are observed (<0.5% between model outputs with predicted and measured NBP concentrations) except for the Rainbow vent sites where our model overestimate NBP substrate concentrations. For this particular Rainbow site, we therefore considered these updated NBP values as boundary conditions when running our microbial model.

Abstract should include rates in volume per year for those that think in those terms as opposed to just focusing on fluxes.

Following reviewer #3’s comment, we added rates in GtC/y. We modified the sentence in the abstract and now state (l. 42-45): “Maximum carbon fixation rates (9 gC m⁻² y⁻¹) could lead to 0.05 GtC y⁻¹ of C biomass produced through chemoorganotrophy within hydrothermal plumes, similar to the Particulate Organic Carbon export fluxes reported in the deep Atlantic and Pacific Ocean.”

Ln 92: $1.16e-8 \text{ s}^{-1}$ – what do these units mean and how was this value chosen?

This value refers to a mortality rate as was stated in the SI of the manuscript (lines 107-118). This mortality rate was used in Reed et al (2015) who performed numerical modelling of bacterial communities in hydrothermal plumes (indicated by the citation 19 in the manuscript). Following their approach, mortality is defined by a first order term for microbial biomass (our state variable). To make it clearer for the reader, we added the following sentence in the SI: “mortality is defined by a first order term for microbial biomass [Ref Reed et al, 2015].”

Response to reviewer #4

Line 94-101: we agree with the reviewer that abiotic processes may generate significant chemical alterations in the nascent buoyant plume, affecting mainly hydrogen, and sulfide and the co-precipitating species (e.g. iron).

In order to emphasize a bit more the chemical alterations that may occur, we modified the sentence and added the two suggested citations. We now state: Non-buoyant plume chemistry is predicted based only on dilution from high temperature fluids with seawater, ignoring thus potential abiotic processes that could occur and alter the chemical distribution in the rising plume^{22,23} (lines 100-101). (...) would only lead to a slight overestimation of substrate concentration and affect only total sulfides and H₂ concentrations²² (line 105).

As stated above, we also performed compared our substrate concentrations in NBP based on our hydrodynamic and dilution only model with chemical data when available and added a dedicated section in the Supplementary info. We then added in the main text : “To prevent any overestimation of substrates in NBP leading to miscellaneous biomass production rates, we compared our NBP predicted values to literature data and corrected them accordingly when needed (see Supplementary Material)” (lines 107-110).

As pointed out by the reviewer, accurate biogeochemical budgets or a detailed description of the biogeochemical processes within the rising plume are beyond the scope of this paper. Besides, the aforementioned chemical alterations would only affect the activity of chemoautotrophic species with little effect on heterotrophic bacteria (except through DOC internal recycling).

Line 110-111: Reference to Reed et al, 2015 was added as requested by reviewer 4. The distribution of the different microbial groups does not refer to Table 1.

Table: As requested by reviewer 4, we added references McNichol et al, 2016 (#56), and Lin et al, 2016 (#57)

Line 153-155: As spotted by reviewer 4, there was an error in the citation index: it was meant to be McNichol et al 2018 (i.e. ref. 14) and not Dick et al 2013 (i.e. ref. 13). Citation was corrected and we now cite Mc Nichol et al, 2018.

SI. Table 1. Following reviewer 4's comment, DOC concentrations, and the repartition between the labile and refractory parts are detailed in Table 1. We now state: "DOC concentrations in TAG fluids were measured from samples collected during the BICOSE and BICOSE2 cruises⁴³. DOC concentrations in hydrothermal fluids from the Logatchev vent sites are set according to⁴⁴. Since no actual data from the Rainbow vent site were available, DOC concentrations of the Rainbow fluids were set according to 100µM, similar to the ultramafic Lost City site as measured by⁴⁵. All other vent sites along the Mid-Atlantic Ridge, and at the Karei and Edmonds vent fields were set to 200 µM^{44,46}. DOC concentrations at the Crab Spa vent site were assumed to be lower^{47,48} and set to 100µM, while DOC concentrations at the Dante and Grand Bonum site were assumed to be around 140µM^{46,49}. All DOC in the hydrothermal fluid were assumed to be labile DOC, as high volatiles and bioavailable compounds concentrations have been documented in hydrothermal fluids^{43,45,47}. Seawater DOC concentration was taken from⁴⁵ and assumed to be mostly refractory^{50,51}, with only 0.01µM of labile DOC⁵²." Thanks to reviewer 4's comment, we adjusted more precisely the DOC concentrations at the different study sites and deep seawater, re-run all simulations and updated the figures accordingly (hence the changes in Table 2).

REVIEWERS' COMMENTS

Reviewer #2 (Remarks to the Author):

The revised manuscript discusses how carbon is being cycled in hydrothermal systems. It also sheds new light on how different microbial communities play a key role in cycling C at different points in the evolution of a hydrothermal plume. For these reasons, I consider this a significant contribution to the field and am recommending this manuscript for publication. I have included one minor edit below to help with clarity for readers.

For figure 2b, please identify the vent site each circle corresponds to in either the caption or image.

Reviewer #3 (Remarks to the Author):

Comments were adequately addressed and effort to address them is appreciated. A nice contribution!

CATHALOT Cécile
Laboratoire des Cycles Géochimiques et Ressources
IFREMER
CS10070
F-29280 Plouzané
France
Tel: 02 98 22 47 54
Email: cecile.cathalot@ifremer.fr

Brest, September 11th, 2021

Subject: Point-by-point Response to reviewer of manuscript NCOMMS-20-39441-T

Dear Dr. Frishkorn,

Please find below a point-by-point response to the reviewer 2's comments.

Please do not hesitate to get back at us if anything is unclear, or if you feel like some explanations need to be deepened, or changes improved.

Thank you again for considering our manuscript for publication in Nature Communications.

Best Regards,

Cécile Cathalot, on behalf of all co-authors

Response to reviewer #2

Figure 2b: The name of each vent site has been added on panel b of figure 2.